# Design and Investigation of One-Handed Interaction Techniques for Single-Touch Gestures

Category: Research

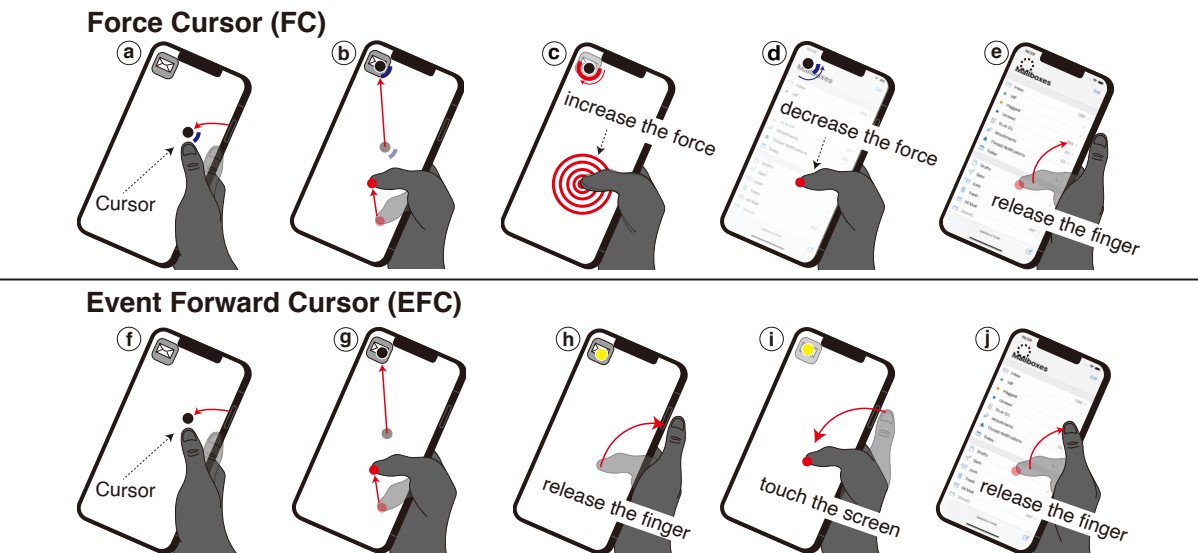

Figure 1: Our two techniques. In Force Cursor (FC, top), the cursor mode is triggered by a user's swipe from the bezel (a); then, the user moves the cursor by dragging the finger (b); a touch-down event is issued at the cursor position when the user increases the force (c), and a touch-up event is issued when decreasing the force (d); the cursor mode ends when the user releases the finger from the screen (e). In Event Forward Cursor (EFC, bottom), the trigger of the cursor mode (f) and the method of moving the cursor (g) are the same as in FC (a and b, respectively); when the finger is released from the screen in the cursor mode (h), the mode switches to the forward mode, and then, when the user performs a gesture anywhere on the screen in the forward mode, the gesture is forwarded to the cursor position; in this figure, a tap, i.e., a touch-down event (i) and touch-up event (j) is forwarded; although the forward mode ends when the user releases the finger from the screen, to enable single-touch gestures that combine touch-up events such as double-tap or tap-and-hold, if the screen is touched again within the double-tap waiting time (0.25 seconds) after the user releases the finger from the screen, the forward mode continues.

## ABSTRACT

Many one-handed interaction techniques have been proposed to interact with a smartphone with only one hand. However, these techniques are all designed for selecting (tapping) unreachable targets, and their performance of other single-touch gestures such as a double-tap, swipe, and drag has not been investigated. In our research, we design two one-handed interaction techniques, Force Cursor (FC) and Event Forward Cursor (EFC), each of which enables a user to perform all single-touch gestures. FC is a cursor technique that enables a user to issue touch events using force; EFC is a cursor technique that enables a user to issue touch events by a two-step operation. We conducted a user study to investigate single-touch gesture performance of one-handed interaction techniques: FC, EFC, and the contents shrinking technique. The result shows that the success rate of one-handed interaction techniques varies depending on the gesture and that EFC has a high success rate independently of the gestures. These results clarified the importance of investigating the performance of single-touch gestures of one-handed interaction techniques.

**Index Terms:** Human-centered computing—Interaction techniques; Human-centered computing—Gestural input

## 1 INTRODUCTION

Many users tend to use only one hand to interact with their smartphones (i.e., holding a smartphone in one hand and using only the thumb) [23, 37]. The possible reason for this is the fact that only the other hand can be used when a user holds an umbrella or baggage in one hand [22]. However, when interacting with the smartphone with only one hand, it is difficult for a user to reach the thumb to all parts of the smartphone's screen without changing the grasping posture because the thumb's reach is limited [5, 32]. Therefore, the user needs to interact with the smartphone while changing the grasping posture as appropriate. However, changing the grasping posture makes the user's grasp of the smartphone unstable, which hinders the user's comfortable interaction with the smartphone [12, 13] and may cause the device to fall.

This problem has been well known in the HCI field; thus, to enable one-handed interaction on a smartphone without changing the grasping posture, HCI researchers have proposed many one-handed interaction techniques (e.g., [8, 25, 28]). However, most of them are designed to allow a user to *select (tap)* unreachable targets. Therefore, while they have shown that these techniques improve a user's target selection (tap) performance compared to the condition without one-handed interaction techniques, the performance of other single-touch gestures (e.g., a double-tap, swipe, drag) have not been investigated. Also, many techniques do not support single-touch gestures other than a tap. Note that these single-touch gestures, including a tap, are frequently used when a user uses a smartphone [3]. For example, a double-tap is used for the skip function in video viewing applications; swipes from the bottom and top bezels are used to return to the home screen and to access an informa-

tion center, respectively; a drag is used for moving an icon on the home screen and selecting multiple photos. Thus, enabling a user to perform such single-touch gestures is important in a one-handed interaction technique. Furthermore, it is important to investigate the single-touch gesture performance of the one-handed interaction technique.

In our research, we designed two techniques, *Force Cursor* (FC, Figure 1 top) and *Event Forward Cursor* (EFC, Figure 1 bottom), for enabling a user to perform single-touch gestures on an unreachable area on a smartphone. Both techniques use a cursor for allowing a user to interact with the smartphone while keeping a stable grasp since the technique using a cursor can stabilize the grasp of the smartphone [9]. The reason why single-touch gestures cannot be performed in previous one-handed interaction techniques using a cursor (hereafter cursor techniques) is that they are designed so that a tap is performed at the cursor position when the user releases the finger from the screen. By contrast, we designed our cursor techniques so that a user can issue touch events (i.e., touch-down events, touch-move events, and touch-up events) at the cursor position. Specifically, FC is a cursor technique that allows the user to issue touch events by changing the force (Figure 1 top). On the other hand, EFC is a cursor technique that involves two steps of operation; the first is for determining the touch event position; the second is for performing single-touch gestures (Figure 1 bottom).

Moreover, to investigate the single-touch gesture performance of one-handed interaction techniques, we conducted a user study with four techniques: our two techniques (FC and EFC), a content shrinking technique (One-handed Mode [41] (OM), which is a technique that allows a user to perform all single-touch gestures), and a direct thumb-touch technique, where a user uses a smartphone without any one-handed interaction technique. Based on the results of the user study, we discuss the performance of each one-handed interaction technique.

The main contributions of this paper are the followings: 1) the design of two one-handed interaction techniques (FC and EFC), each of which enables a user to perform all single-touch gestures while keeping the grasp of the smartphone stable, and 2) the single-touch gesture performance of one-handed interaction techniques, which has not been investigated before. The results show that OM is a fast technique for performing gestures, FC is a technique that can stabilize the user's grasp of the smartphone, and EFC is a technique with a high success rate regardless of the gestures to be performed. It is also found that the success rate in OM and FC varies greatly depending on the gestures to be performed, indicating the importance of investigating the single-touch gesture performance.

## 2 RELATED WORK

Many one-handed interaction techniques have been proposed to facilitate one-handed interaction on smartphones. We categorize these techniques into three: screen transformation techniques, proxy region techniques, and cursor techniques.

### 2.1 Screen Transformation Techniques

These techniques enable a user to move the contents shown on the screen (hereafter the contents), shrink the contents, or both.

On the iPhone 6 and later, a function called "Reachability" has been introduced [1]. With Reachability, a user can move the contents down by double-tapping the home button or swiping down on the bottom edge of the screen. On the iPhone 6 and later, a user can move the contents down by double-tapping the home button or swiping down on the bottom edge of the screen. Similarly, Palm-Touch [31] can be used to move the contents down by touching the screen with the palm. In Telekinetic Thumb [20], a user can move the contents to the lower right of the screen by performing a pull gesture above the screen. Sliding Screen [25] is triggered by a swipe from the smartphone's bezel or a touch with a large

contact area; when a user drags the thumb, the contents are moved point-symmetrically or in the direction of the thumb's movement. MovingScreen [45] is similarly designed to move the contents in a point-symmetrical manner in the direction of the thumb's movement and is triggered by a swipe from the bezel; it differs in that the movement's speed changes in proportionally to the swipe distance on the bezel where it is triggered. TiltSlide [8] is also a technique to move the contents in the same way and triggered by tilting the smartphone. In IndexAccess [18] and the technique proposed by Le et al. [30], a touchpad is attached to the back of the smartphone; the index finger's movement on the touchpad moves the contents. In these techniques, after the contents are moved, a part of the contents is hidden outside the screen; and thus, dragging a content (e.g., icon) from the area beyond the reach of the thumb to the hidden area is not supported, and vice versa.

Galaxy's One-handed Mode [41] (OM) is triggered by a triple-tap of the home button or a swipe from the four corners of the screen. It shrinks the contents and places them near the thumb; by default, the contents shrink to two-thirds of their size. Similarly, TiltReduction [41] is also a technique to shrink the contents and triggered by tilting the smartphone. These techniques that shrink the contents can be adapted so that a user can perform all single-touch gestures, although the single-touch gesture performance has not yet been investigated. Thus, we investigate the performance.

### 2.2 Proxy Region Techniques

In the proxy region techniques, a user can use a different area of the screen or around the screen as an alternative area to operate the unreachable area.

ThumbSpace [24] is triggered by a drag on the screen; it then displays a popup view that miniaturized all contents of the screen. Although this technique solves the problem that the thumb cannot reach all parts of the smartphone's screen without changing the grasping posture, it reduces the size of any target because it shrinks the all contents of the screen, which may lead to the fat finger problem [43] and occlusion (i.e., a small target is occluded by the thumb). TapTap [40] is triggered when a user touches the screen; it then displays a popup on the center of the screen, which shows an enlarged view of the area around the touched position on the screen. However, it is difficult to use this technique for an area too far from the thumb's reach because the user needs to touch around the area that the user wants to interact with. Hasan et al. [15] proposed the technique that uses the in-air space above the screen, which allows the user to interact with unreachable targets by using the three-dimensional movements of the thumb.

In the technique proposed by Lochtefeld [34], a touchpad is attached to the back of a smartphone, which allows a user to operate an unreachable target by touching the target from the back of the device using the index finger; this design utilizes the fact that the reachable area with one-handed interaction can be extended by 15 percent by using the index finger on the back of the smartphone [52]. However, these techniques [15, 34] require an additional device.

### 2.3 Cursor Techniques

In cursor techniques, a user can use the cursor for selecting an unreachable target instead of touching the target directly.

Most of the previous cursor techniques switch to the mode to use the cursor (cursor mode) when a user performs a predetermined gesture as a trigger, and then the cursor appears under the finger, and the user can move the cursor according to the distance of the finger movement by dragging the finger while in the cursor mode. For example, TiltCursor [8] is triggered by tilting the smartphone, BezelCursor [33] is triggered by a swipe from the bezel, Extendible Cursor [25] is triggered by a swipe from the bezel or a touch with

a large contact area, ExtendedThumb [28] is triggered by a double-tap, and MagStick [40] is triggered by touching the screen.

Unlike these techniques, in CornerSpace and BezelSpace [53], the cursor appears in places other than under the finger. CornerSpace [53] displays the popup of shrunk all contents of the screen when a user swipes from the bezel. Then, when the user touches on the popup, the cursor appears at the position corresponding to the touched position on the screen. On the other hand, in BezelSpace [53], when a user swipes from the bezel, buttons representing the four corners and center of the screen appears, and then the cursor appears at the corresponding position when the user presses the button.

2D-Dragger [44], ForceRay [9], and HeadReach [46] are techniques that use a different method of moving the cursor. In 2D-Dragger [44], when a user moves the finger, the cursor is moved to the nearest target in the direction of the finger's movement. In ForceRay [9], the cursor moves away from the finger when a user increases the force, and the cursor moves in the direction of the finger when the force is decreased. In HeadReach [46], a method of moving the cursor combining the direction of the face and dragging a finger is used.

Although the triggers, the initial cursor position, and the method of moving the cursor are different in these previous cursor techniques, the mechanism to issue touch events (i.e., selecting target) is the same in all of them: when a user releases the finger from the screen during the cursor mode, a tap is performed (i.e., both touch-down and touch-up events are issued simultaneously) at the cursor position. This approach makes it impossible for a user to perform single-touch gestures other than a tap. On the other hand, our two techniques (FC and EFC) enable a user to perform all single-touch gestures using a cursor. [14] is a previous study of the same concept as ours. However, no experiments were conducted with participants other than the authors.

## 3 DESIGN OF OUR TECHNIQUES

We designed two one-handed interaction techniques that enable a user to perform all single-touch gestures using a cursor. Since previous studies [8, 9, 25] show the high performance of the cursor technique triggered by a swipe from the bezel, we adopted a swipe from the bezel as the trigger to switch to the cursor mode in both techniques (Figure 1a, f). Moreover, we designed both techniques so that the cursor moves in the same direction as the finger movement (Figure 1b, g), as in [8, 33].

### 3.1 Force Cursor (FC)

In FC, the following operations are required to issue touch events at the cursor position. Firstly, the user performs a swipe from the bezel to switch to the cursor mode (Figure 1a), then, the user drags the finger to move the cursor to the desired position (Figure 1b); the cursor movement distance is calculated by multiplying the thumb movement distance by the control-display ratio.

A touch-down event is issued when the user increases the force above a threshold (Figure 1c). A touch-up event is issued when the user decreases the force below the threshold after the touch-down event is issued. The cursor mode continues until the user releases the finger from the screen, allowing the user to continuously use the cursor to perform a gesture to an unreachable area. Since this design allows the user to issue touch-down, touch-move, and touch-up events by controlling the force, all single-touch gestures can be performed. For example, a tap is performed by increasing the force and then decreasing the force (i.e., clicking using force [51]), a double-tap is performed by quickly repeating the click twice, and a swipe or drag is performed by moving the finger while the force is applied.

We conducted a demonstration to confirm if users can use FC and found the following problems with FC based on the comments from the demonstration participants. Firstly, it is not possible to know

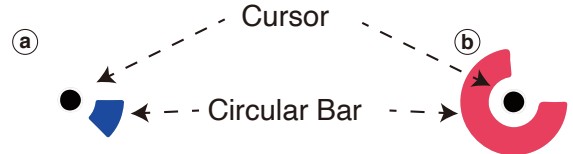

Figure 2: Circular bar that represents the current applied force displayed around the cursor of FC. The bar is blue when the force is below the threshold, and red when it is above.

how much force the user is currently applying without feedback. Secondly, it is difficult to perform double-tap using a cursor with force (i.e., to quickly repeat a click using force twice). Thirdly, it is difficult to make large movements of the finger while applying high force (i.e., performing a drag using the cursor). Therefore, we added the following three functions to solve these problems.

#### 3.1.1 Visual Force Feedback

Previous studies [7, 35, 48] have shown that continuous visual feedback is effective in a technique using force. Therefore, we display a circular bar to provide continuous visual feedback to a user (Figure 2). The circular bar is displayed in blue when the current applied force is below the threshold (Figure 2a) and in red when it is above the threshold (Figure 2b).

#### 3.1.2 Double-Tap Assistance

In the demonstration, the reason why novices often failed to double-tap using FC was that they changed the force before the force threshold was crossed, i.e., before the touch-up event of the first tap was issued, the user was attempting to issue a touch-down event for the second tap. Therefore, we implemented a function to enable a user to perform a double-tap using the cursor without being aware of the threshold. In this function, a user first increases the force above the threshold, then decreases the force, increases it again, and finally decreases the force below the threshold to perform a double tap (i.e., the user only needs to cross the force threshold when increasing the force to issue the touch-down event of the first tap and when decreasing the force to issue the touch-up event of the second tap).

#### 3.1.3 Drag Assistance

It is difficult to move a finger with increased force because of the increased frictional force between the finger and the screen [16, 17]. In order to solve this problem, we implemented a function that, when the force is increased to the maximum detectable value and dwelled for 1.0 seconds, then, a touch-move event is issued continuously at the cursor position, independent of the force, until the finger is released from the screen. Therefore, the user can perform a drag using the cursor with low force; this function of fixing the force state is the same as that of *force lock* [17].

### 3.2 Event Forward Cursor (EFC)

In EFC, in the same way as in FC, when a user performs a swipe from the bezel, the cursor mode is triggered (Figure 1f), and then, the user drags the finger to move the cursor to the desired location (Figure 1g). In EFC, when the user releases the finger from the screen during the cursor mode, the mode switches to *forward mode* (Figure 1h); the cursor is yellow while in the forward mode so that the user knows the current mode. While in the forward mode, all touch events are forwarded to the cursor position. That is, touch-down events are issued at the cursor position when the finger touches the screen, then touch-move events are issued at the cursor position until the user releases the finger from the screen, and

touch-up events are issued at the cursor position when the user releases the finger from the screen. Although the forward mode ends when the user releases the finger from the screen, to enable single-touch gestures that combine touch-up events such as double-tap or tap-and-hold, if the screen is touched again within the double-tap waiting time (0.25 seconds) after the user releases the finger from the screen, the forward mode continues.

## 4 USER STUDY: INVESTIGATION OF THE PERFORMANCE OF SINGLE-TOUCH GESTURES

To investigate the single-touch gesture performance of one-handed interaction techniques, we conducted a user study with 8 participants (21 – 24 years old, M = 22.38, SD = 1.06; 7 males; who have the iPhone that can detect force; and usually interact with their smartphones with the right hand).

For the safety of the participants, we conducted this user study remotely via video call.

### 4.1 Setup

#### 4.1.1 Apparatus

Participants used their own iPhones; the iPhone's force sensitivity is set to 'firm'. Three iPhone XS and one iPhone X (screen size: 135.10 mm × 62.39 mm), and one iPhone 8, two iPhone 7, and one iPhone 6s (screen size: 103.94 mm × 58.44 mm) were used in this user study. Since the implementation of the experimental application uses "pt" as a unit of length and the actual size of a pt varies slightly depending on the iPhone, in this section, we use pt as a unit of length. In the case of the iPhone XS and iPhone X, the 1 pt $\simeq$ 0.17 mm, and in the case of the iPhone 8, iPhone 7 and iPhone 6s, the 1 pt $\simeq$ 0.16 mm.

#### 4.1.2 Techniques

We used FC, EFC, and One-Handed Mode (OM, same as Samsung's [41]) as one-handed interaction techniques that enable a user to perform all single-touch gestures. Other one-handed interaction techniques, such as Apple's Reachability [1] or cursor techniques (e.g., [25, 33]), were excluded from this user study because they do not support all single-touch gestures. In addition, as a baseline, we used Direct Touch (DT), i.e., without a one-handed interaction technique. That is, there are four different techniques (DT, OM, FC, and EFC) used in this user study.

We unified triggers of OM, FC, and EFC to the swipe from the bezel to avoid the effect of different triggers. For FC and EFC, the size of the cursors was 9 pt, and the cursor's control-display ratio was set to be three times, that is, the cursor moves three times the distance of the finger movement.

In One-Handed Mode (OM), when a user swipes from the bezel, the contents are shrunk; swipe again to restore the contents to their original size. In this user study, the contents is shrunk to 2/3 of its original size and moved to the lower right corner of the screen (Figure 3e), just like the standard Galaxy setting.

For our implementation of Force Cursor (FC), we used the force readings provided by the iPhone's force-sensitive touchscreen. According to Apple's documentation [2], with the force sensitivity set to "firm", force is capable of measuring with unitless value from 0 to $\frac{480}{72}$ ($\simeq$ 4.0 N [36]); with values around 1.0 ($\simeq$ 0.60 N [36]) being the force applied by ordinary touch. We set the force threshold for issuing touch-down and touch-up events in FC to 3.0 ($\simeq$ 1.8 N).

#### 4.1.3 Targets

18 targets were placed on an invisible 15 × 7 grid, as shown in Figure 3a; the size of the grid varies with the size of the screen. The target size was set to two different values: 60 pt × 60 pt (*Large*, Figure 3a, b, d) and 30 pt × 30 pt (*Small*, Figure 3c). The target was placed at the top-left corner of the grid as a starting point. In addition, if part of the target protrudes from the screen, the target

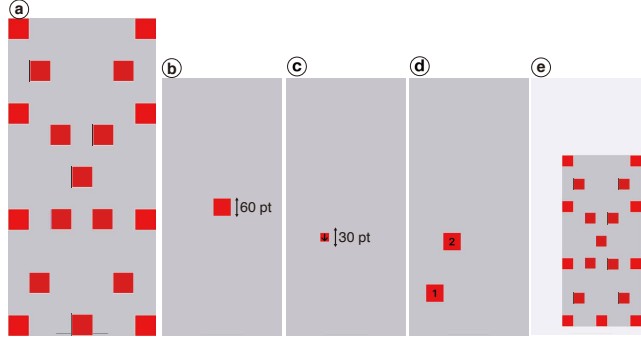

Figure 3: Targets and screenshots of the user study. a: 18 targets placed on the screen. b: A target for tap and double-tap sessions; the target size is Large. c: A target for swipe sessions; the target size is Small and the swipe direction is down. d: A target for drag sessions; the target size is Large. e: The screen when OH is used; the contents are shrunk to 2/3 of its original size.

was moved to the center of the screen by the amount of the protrusion. During the task, only the current target is displayed in red, and other targets are not displayed (Figure 3b, c, d).

The location and size of the target were based on the experiments of [9]. However, while [9] did not place targets at hand, we place targets on the entire screen (include at hand) since different sizes of smartphones change the unreachable area of the user [32].

#### 4.1.4 Single-Touch Gestures

We used the commonly used four single-touch gestures: tap (*Tap*), swipe (*Swipe*), double-tap (*DTap*), and drag (*Drag*). We set up a session for each gesture.

In a session of Tap, a target is displayed in red (Figure 3b) and participants perform a tap on a target. The tap is performed when both the touch-down event and the touch-up event are issued on the same target.

In a session of DTap, a target is displayed in red (Figure 3b), participants perform a double-tap on the displayed target. A double-tap is a gesture that the user touches the same target again within the double-tap waiting time after performing a tap and then releases the finger from the screen on the target. In this user study, the waiting time of the double-tap was set to 0.25 seconds: this is the same as the default of the iPhone.

In a session of Swipe, participants swipe the target in the direction of the arrow displayed on the target (Figure 3c). The direction of a swipe was randomly selected from four directions, up, down, right, and left when the target was updated. However, because a swipe toward the screen bezel on the target that is in contact with the screen bezel cannot be recognized, directions toward the screen bezel were removed from the selection (e.g., one direction was randomly selected from the top, left, and bottom directions for the rightmost target).

In a session of Drag, two targets were displayed (Figure 3d), and participants dragged a target labeled '1' (target1) to a target labeled '2' (target2). Target2 was randomly selected from 17 targets other than target1.

### 4.2 Task and Procedure

Before starting this user study, we initiated a video call with the participants to explain this user study. Participants sat on a chair and interacted with their iPhones with their right hands and only use the thumb.

We recorded 4 techniques × 4 gestures × 18 targets × 2 target sizes × 2 repetitions = 1,152 trials per participant.

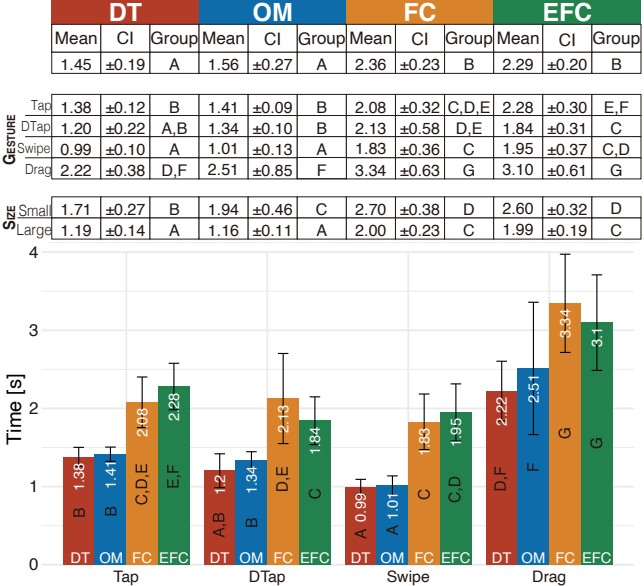

Figure 4: The results of Time [s]. The top is a table summarizing the results of Technique, Technique × Gesture, and Technique × Size, and the bottom is a bar graph showing the results of Technique × Gesture. Pairs that do not share a letter of Group are significantly different. Whiskers denote 95% CI.

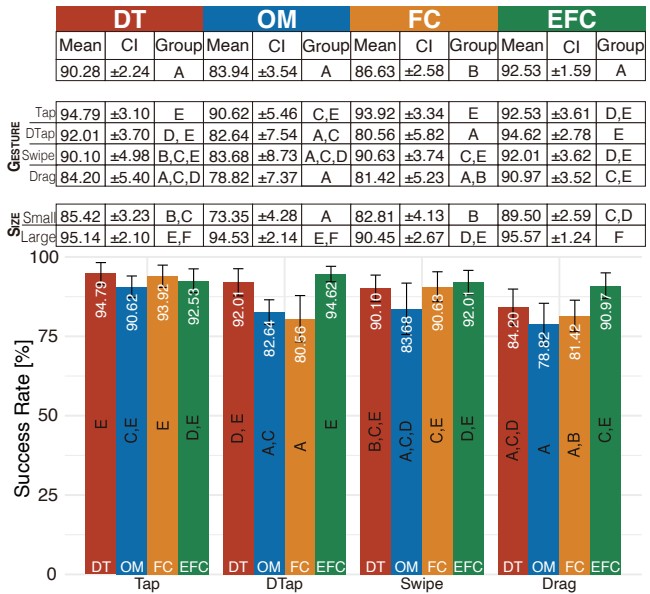

Figure 5: The results of Accuracy [%]. The top is a table summarizing the results of Technique, Technique × Gesture, and Technique × Size, and the bottom is a bar graph showing the results of Technique × Gesture. Pairs that do not share a letter of Group are significantly different. Whiskers denote 95% CI.

The techniques and gestures were counterbalanced using a Latin Square. The display order of the targets was random. Target size was also selected in random order, however, out of the 16 combinations of the 4 techniques × 4 gestures, 8 of the 16 started with small targets, and the remaining 8 with large targets. Before starting a new gesture session with each technique, participants performed the gesture to 18 targets as a practice. Then, the gesture was performed to 18 targets, repeated twice at the same size, and repeated twice again at the other size. Successful or unsuccessful performance of the gesture was informed to the participant in different sounds. When the gesture was successfully performed, the next target was displayed, and when the gesture failed, the same target was displayed again.

After the completion of the 2 × 18 trials at both sizes, participants began the next gesture session. Then, after 4 gesture sessions were completed, the next technique was presented to the participants. To know the subjective evaluation, we asked participants to answer the questionnaire of System Usability Scale [6] (SUS) for the technique after all gesture sessions for the technique were completed.

The participants took breaks as needed to avoid hand fatigue. The user study took about two hours, and the participants received $33.1 as a reward.

## 4.3 Result

The independent variables were Technique (DT, OM, FC, and EFC), Gesture (Tap, DTap, Swipe, and Drag), and Size (Large and Small). The dependent variables are trial completion time (Time), success rate (Accuracy), the jerk (Jerk) and angular acceleration (Angular Acceleration) used to evaluate the stability of the smartphone, and the score of SUS (SUS Score). Jerk is used to evaluate smooth motion [29] and angular acceleration is used to evaluate the vibration state [54]. For the analysis, we used a repeated-measures ANOVAs. Since the purpose of this user study is to investigate the single-touch gesture performance of one-handed interaction techniques, we only describe the main effect of

the Technique and the related interaction effects.

Because this user study conducted remotely and participants used their own smartphones, there were two different sizes of smartphones. However, there were no significant effect of smartphone size (Time: $F(1,6) = 3.35$, p = 0.12; Accuracy: $F(1,6) = 2.86$, p = 0.14; Jerk: $F(1,6) = 0.14$, p = 0.72; Angular Acceleration: $F(1,6) = 2.23$, p = 0.19).

### 4.3.1 Time

Time results are shown in Figure 4. The reason why Tap was slower than DTap and Swipe is that there was a waiting time of 0.25 seconds to judge whether a double-tap is performed or not after a tap is performed. On the other hand, a double-tap is confirmed immediately after the second tap is performed. Inherently, a tap takes longer than a double-tap [21].

Technique had a significant main effects on Time ($F_{3,9425} = 25.05, p < .001$). Tukey's HSD test was also significant (p < .05 between OM and FC, p < .01 between DT and FC, and others p < .001). There was also significant Technique × Gesture interaction effect ($F_{9,9425} = 3.26, p < .01$). Tukey's HSD test was also significant (p < .05 between OM's DTap and OM's Swipe, OM's Drag and FC's DTap, FC's DTap and EFC's DTap, and EFC's Tap and EFC's Swipe, p < .01 between DT's Tap and DT's Swipe, DT's Tap and OM's Swipe, and DT's Swipe and OM's DTap, DT's Drag and FC's Swipe, DT's Drag and EFC's DTap, OM's Tap and OM's Swipe, and FC's DTap and FC's Swipe, and others p < .001). As shown Figure 4, for all gestures, Time was DT ≃ OM < EFC ≤ FC. In other words, the technique of touching the target directly with a finger is faster than the technique of operating the target indirectly with a cursor; this is similar to the results shown by Chang et al. [8] in a tap-only experiment. In addition, there was also significant Technique × Size interaction effect ($F_{3,9425} = 3.52, p < .05$). Tukey's HSD test was also significant (p < .01 between DT's Small and OM's Small, and others p < .001). As expected, in Time, Large was faster than Small for all Technique.

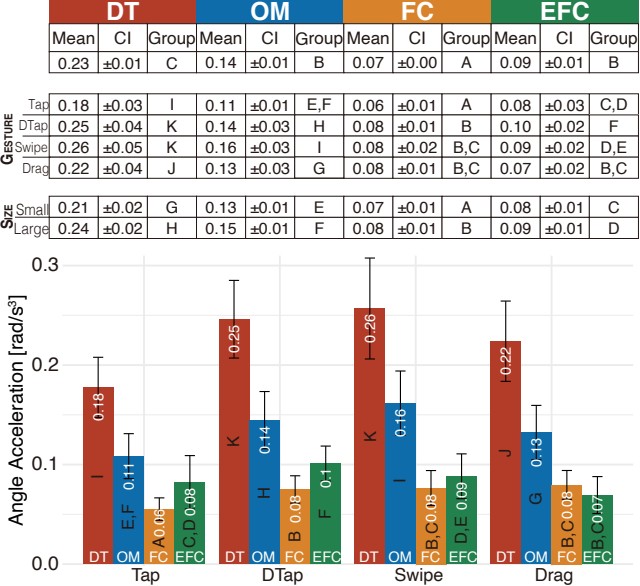

| | | DT | | | OM | | | FC | | | EFC | |
|---|---|---|---|---|---|---|---|---|---|---|---|---|
| | Mean | CI | Group | Mean | CI | Group | Mean | CI | Group | Mean | CI | Group |
| | 0.23 | ±0.01 | C | 0.14 | ±0.01 | B | 0.07 | ±0.00 | A | 0.09 | ±0.01 | B |
| **GESTURE** Tap | 0.18 | ±0.03 | I | 0.11 | ±0.01 | E,F | 0.06 | ±0.01 | A | 0.08 | ±0.03 | C,D |
| DTap | 0.25 | ±0.04 | K | 0.14 | ±0.03 | H | 0.08 | ±0.01 | B | 0.10 | ±0.02 | F |
| Swipe | 0.26 | ±0.05 | K | 0.16 | ±0.03 | I | 0.08 | ±0.02 | B,C | 0.09 | ±0.02 | D,E |
| Drag | 0.22 | ±0.04 | J | 0.13 | ±0.03 | G | 0.08 | ±0.01 | B,C | 0.07 | ±0.02 | B,C |
| **SIZE** Small | 0.21 | ±0.02 | G | 0.13 | ±0.01 | E | 0.07 | ±0.01 | A | 0.08 | ±0.01 | C |
| Large | 0.24 | ±0.02 | H | 0.15 | ±0.01 | F | 0.08 | ±0.01 | B | 0.09 | ±0.01 | D |

**Figure 6:** The results of Jerk [$\frac{m}{s^3}$]. The top is a table summarizing the results of TECHNIQUE, TECHNIQUE × GESTURE, and TECHNIQUE × SIZE, and the bottom is a bar graph showing the results of TECHNIQUE × GESTURE. Pairs that do not share a letter of Group are significantly different. Whiskers denote 95% CI.

| | | DT | | | OM | | | FC | | | EFC | |
|---|---|---|---|---|---|---|---|---|---|---|---|---|
| | Mean | CI | Group | Mean | CI | Group | Mean | CI | Group | Mean | CI | Group |
| | 0.46 | ±0.04 | C | 0.29 | ±0.03 | B | 0.17 | ±0.01 | A | 0.21 | ±0.02 | B |
| **GESTURE** Tap | 0.36 | ±0.07 | I | 0.23 | ±0.06 | E | 0.13 | ±0.03 | A | 0.19 | ±0.06 | C,D |
| DTap | 0.51 | ±0.11 | K | 0.32 | ±0.09 | G | 0.17 | ±0.04 | B,C | 0.24 | ±0.05 | E |
| Swipe | 0.53 | ±0.12 | L | 0.34 | ±0.09 | H | 0.18 | ±0.05 | B,C | 0.21 | ±0.06 | D |
| Drag | 0.45 | ±0.10 | J | 0.27 | ±0.08 | F | 0.19 | ±0.05 | B,D | 0.18 | ±0.06 | B |
| **SIZE** Small | 0.44 | ±0.04 | F | 0.27 | ±0.04 | D | 0.15 | ±0.02 | A | 0.20 | ±0.02 | B,C |
| Large | 0.49 | ±0.06 | G | 0.32 | ±0.06 | E | 0.18 | ±0.02 | B | 0.22 | ±0.03 | C |

**Figure 7:** The results of Angular Acceleration [$\frac{rad}{s^2}$]. The top is a table summarizing the results of TECHNIQUE, TECHNIQUE × GESTURE, and TECHNIQUE × SIZE, and the bottom is a bar graph showing the results of TECHNIQUE × GESTURE. Pairs that do not share a letter of Group are significantly different. Whiskers denote 95% CI.

#### 4.3.2 Accuracy

Accuracy results are shown in Figure 5. TECHNIQUE had a significant main effects on Accuracy ($F_{3,473} = 8.24, p < .001$). Tukey's HSD test was significant (p < .05 between OM and FC; p < .01 between DT and FC; and p < .001 between FC and EFC). There was also significant TECHNIQUE × GESTURE interaction effect ($F_{9,473} = 5.92, p < .001$). Tukey's HSD test was also significant (p < .05 between DT 's DTap and OM's DTap, DT's Swipe and FC's DTap, DT's Drag and FC's Tap, OM's Tap and FC's DTap, OM's Tap and FC's Drag, OM's DTap and EFC's Tap, OM's DTap and EFC's Swipe, OM's Swipe and FC's Tap, FC's DTap and FC's Swipe, FC's Swipe and FC's Drag, and FC's Drag and EFC's Drag; p < .01 between DT's Tap and DT's Drag, DT's Tap and OM's Swipe, DT's DTap and FC's DTap, DT's DTap and FC's Drag, DT's Swipe and OM's Drag, DT's Drag and EFC's DTap, OM's DTap and FC's Tap, OM's Swipe and EFC's DTap, FC's DTap and EFC's Drag, FC's DTap and EFC's Drag, FC's DTap and EFC's Swipe, and FC's Drag and EFC's Swipe, and FC's Drag and EFC's Tap; and others p < .001). As shown in Figure 5, there was no significant difference between TECHNIQUE in Tap. However, there were significant differences across TECHNIQUE in other gestures. Although EFC had a higher success rate for all gestures, OM and FC had a lower success rate for DTap and Drag. In addition, there was also significant TECHNIQUE × SIZE interaction effect ($F_{3,473} = 24.62, p < .001$). Tukey's HSD test was also significant (p < .05 between DT's Small and FC's Large, DT's Large and EFC's Small, OM's Large and EFC's Small, and FC's Large and EFC's Large; p < .01 between FC's Small and EFC's Small, and EFC's Small and EFC's Large; and others p < .001). As expected, Large had a higher success rate than Small for all techniques. The difference in success rates between SIZE (Large and Small) was smaller for techniques using a cursor (FC: 7.64%, EFC: 6.07%) and larger for other two techniques (DT: 9.72%, OM: 21.18%); this result may be due to the fact that the techniques of touching the target directly with a finger are susceptible to the fat finger problem [43]

and occlusion.

#### 4.3.3 Stability of the Smartphone (Jerk and Angular Acceleration)

Jerk results are shown in Figure 6. TECHNIQUE had a significant main effects on Jerk ($F_{3,9425} = 64.06, p < .001$). Tukey's HSD test was significant (p < .05 between FC and EFC; and others p < .001). There was also significant TECHNIQUE × GESTURE interaction effect ($F_{9,9425} = 51.91, p < .001$). Tukey's HSD test was also significant (p < .05 between OM's DTap and OM's Swipe, FC's DTap and EFC's Tap; p < .01 between FC's Drag and EFC's Swipe, EFC's Swipe and EFC's DTap; and others p < .001). In Jerk, FC < EFC < OM < DT for all gestures except Drag, and FC ≃ EFC < OM < DT for Drag. In addition, there was also significant TECHNIQUE × SIZE interaction effect ($F_{3,9425} = 13.44, p < .001$). Tukey's HSD test was also significant (p < .01 between FC's Large and EFC's Small, EFC's Small and EFC's Large; and others p< .001). In all techniques, Large had significantly lower jerk than Small.

Angular Acceleration results are shown in Figure 7. TECHNIQUE also had a significant main effects on Angular Acceleration ($F_{3,9425} = 37.46, p < .001$). Tukey's HSD test was also significant (p < .05 between FC and EFC; and others p < .001). There was also significant TECHNIQUE × GESTURE interaction effect ($F_{9,9425} = 56.12, p < .001$). Tukey's HSD test was also significant (p < .05 between DT's DTap and DT's Swipe, FC's Swipe and EFC's Swipe; p < .01 between OM's DTap and OM's Drag, FC's DTap and EFC's Swipe, EFC's Tap and EFC's Drag, and EFC's DTap and EFC's Swipe; and others p < .001). As with Jerk, in Angular Acceleration, FC < EFC < OM < DT for all gestures except Drag, and FC ≃ EFC < OM < DT for Drag. In addition, there was also significant TECHNIQUE × SIZE interaction effect ($F_{3,9425} = 23.79, p < .001$). Tukey's HSD test was also significant (p < .01 between FC's Large and EFC's Small, EFC's Small and EFC's Large; and others p< .001). Large had significantly lower angular acceleration than Small for all techniques except EFC.

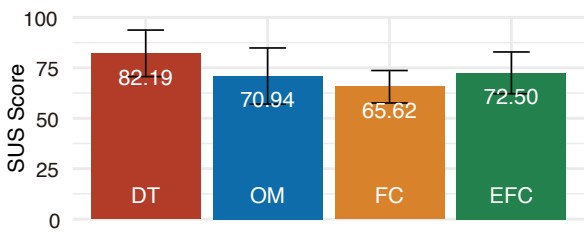

Figure 8: The results of SUS Score. Whiskers denote 95% CI.

Table 1: Time [s] of the trials where the trigger was performed. CI denote 95% CI.

| Gesture | OM | | FC | | EFC | |
|---|---|---|---|---|---|---|
| | Mean | CI | Mean | CI | Mean | CI |
| Tap | 3.82 | 0.41 | 4.14 | 0.67 | 2.28 | 0.24 |
| DTap | 3.81 | 0.59 | 4.01 | 0.62 | 1.84 | 0.25 |
| Swipe | 3.28 | 0.36 | 3.31 | 0.44 | 1.95 | 0.29 |
| Drag | 5.09 | 1.62 | 3.87 | 0.65 | 3.10 | 0.50 |

Table 2: Accuracy [%] of the trials where the trigger was performed. CI denote 95% CI.

| Gesture | OM | | FC | | EFC | |
|---|---|---|---|---|---|---|
| | Mean | CI | Mean | CI | Mean | CI |
| Tap | 85.63 | 4.93 | 80.27 | 3.52 | 92.53 | 2.81 |
| DTap | 73.13 | 4.49 | 68.03 | 5.74 | 94.62 | 2.41 |
| Swipe | 85.00 | 4.32 | 72.56 | 4.54 | 92.01 | 2.78 |
| Drag | 70.98 | 5.19 | 74.59 | 3.07 | 90.97 | 2.77 |

In summary, FC has the smallest movement of smartphones, followed by EFC, OM, DT. The reason for this may be that the time the thumb touches the screen in the techniques using a cursor (FC and EFC) is longer than in the other two techniques (DT and OM). In particular, since the thumb always touches the screen in FC, the smartphone was more stable than in EFC.

### 4.3.4 SUS Score

The result of SUS Score is shown in Figure 8. As a result of ANOVA, SUS Score did not have a significant main effect on TECHNIQUE. In terms of average value, DT was the highest, followed by EFC, OM, then FC. The reason why DT has the highest average value is that SUS has a high score for familiar techniques [42]; DT was the same technique that the participants usually use when using the smartphone and was a familiar technique. Although the average values of OM and EFC were almost the same, the average value of FC was slightly lower than these. This is thought to be because FC was inferior to other techniques in terms of Time and Accuracy.

## 5 DISCUSSIONS

In this section, we discuss the performance of one-handed interaction techniques based on the results of the user study.

### 5.1 The Need to Investigate Single-Touch Gesture Performance

As shown in Figure 4, Figure 6, and Figure 7, the differences in Time, Jerk, and Angular Acceleration across TECHNIQUE did not vary within GESTURE; that is, in Time, DT < OM < FC $\simeq$ EFC, and in Jerk and Angular Acceleration, FC < EFC < OM < DT. However, as shown in Figure 5, although the success rate was higher for all techniques in Tap, the success rates of DT, OM and FC were lower for in the other gestures, depending on the gestures. In addition, there were comments of participants that Drag in OM was difficult, or Swipe was easy but DTap was difficult in FC, or DTap in EFC was easier. These results suggest that the performance of the one-handed interaction technique varies depending on the performed gestures. Therefore, we think that it is important to investigate the single-touch gesture performance of one-handed interaction techniques.

### 5.2 Selecting a Suitable One-handed Interaction Technique

In summary, the results of the user study show that OM is the best technique with a high success rate and fast performing gestures when the target is large and some smartphone movement is allowed. On the other hand, when the target size is small, the success rate of all gestures (especially those other than a tap) is quite low. Although FC had a lower success rate than EFC, the movement of the smartphone was the smallest. Therefore, FC is considered to be a suitable technique for stable the grip of the smartphone. EFC takes more time to perform gestures than OM, however, it has a

high success rate regardless of the target size and gestures, and it can stabilize the grip of the smartphone more than DT and OM. In other words, EFC is considered to be suitable for careful manipulation and for performing gestures on small unreachable targets.

However, the performance of the one-handed interaction technique varies greatly depending on the situation. Therefore, we think that it is important to allow the user to choose the one-handed interaction technique to be used according to the situation, for example, to enable both OM and EFC to be used; the introduction of multiple one-handed interaction techniques can be easily accomplished by assigning a separate trigger to each.

### 5.3 Effects of Number of Performed Triggers

In the user study, all gestures were performed using each technique. However, each technique requires the user to perform trigger at different times: OM allows the user to keep the contents shrunk until the user performs the trigger again; FC allows the user to manipulate the cursor until the user removes the finger from the screen after performing the trigger once; EFC, on the other hand, requires the user to perform the trigger each time a gesture is performed to with the cursor. Therefore, the results of the user study may be influenced by the number of performed trigger.

To analyze the effect of the number of performed trigger, we extract only the trials where one or more triggers were used. The results of Time and Accuracy are shown in Table 1 and Table 2. Jerk and Angular Acceleration did not vary much whether the trigger was performed; in other words, it is almost identical to Figure 6 and Figure 7.

Based on this result, EFC might be the best technique if the user needs to perform the trigger frequently. However, because the number of trials for which the trigger was performed varies greatly depending on the technique (OM: 142 times, FC: 791 times, EFC: 2,304 times), this result may be not robust. In the user study, in OM, participants continued the task with reduced contents after the first trigger was performed. However, given the actual usage environment, we expect the number of the performed trigger to increase because the user resizes the contents to its original size to select a small button, type a letter, or enjoy the displayed content (videos and texts). Therefore, We need to investigate the performance of the technique in real usage environments.

### 5.4 Causes of Gesture Failure

In OM, the error was caused by the difficulty of pointing due to the small size of the target (i.e., fat finger problem [43] and occlusion). Particularly in gestures other than Tap, participants were strongly influenced by small targets. Therefore, the error rate may

be reduced by combining OM and techniques to select small targets [4,38] or techniques to improve the accuracy of touch [19,39].

In FC, many participants said that changing force caused the cursor to move, resulting in an error. In particular, in DTap, the participants had to change the force quickly and repeatedly, which caused the cursor to move, and the success rate was low. In addition, an error occurred because the participants unintentionally applied force when moving the cursor, causing a touch event to issue. These may be improved by introducing a filter that separates cursor movement from changes in force. Corsten et al. [11] found that the performance of the technique using force improves with long-term training, so it is possible that it may improve with more training.

In EFC, participants commented that the cursor was moved unintentionally when they released their finger from the screen to decide where to forward the gesture, resulting in an error. The problem that the touch position changes when the user releases the finger from the screen has been investigated by Xu et al. [50]. Since the cursor was moved three times the distance the finger moved in the user study, the effect of this problem was likely to be stronger. Therefore, in EFC, it may be possible to increase the success rate by taking advantage of the state just before the user releases the finger from the screen, as in [10].

## 6 LIMITATIONS AND FUTURE WORK

In this section, we discuss the limitations of the results of the user study and discuss our future work.

### 6.1 Participants' Individual Attribute and Number of Participants

The participants in the user study were all young and familiar with the interaction with smartphones. However, a wide range of people, from small children to the elderly, novices to experts use smartphones. In particular, elderly people are known to have difficulty with force control [26] and pointing at small targets using their thumbs [27, 49]. That is, age may affect the performance of FC and OH. These are suggesting that the performance of each technique may vary with individual attributes. In addition, because the number of participants is eight, the results of the user study might be not robust. Therefore, we need to conduct an additional user study with more participants of different individual attributes.

### 6.2 Effects of Long-Term Training

In the user study, each participant performed each gesture 108 times in each technique, including practice. However, if the participant trained more, the performance might be change. In particular, Corsten et al. [11] found that the performance of the techniques using force improves with practice. In addition, the performance of other techniques (OM and EFC) might also change as the user becomes an expert. This indicates the need for long-term experiments to accurately determine the performance of one-handed interaction techniques.

### 6.3 Use in Different Environments

We conducted the user study with participants seated in a chair. However, smartphones are used in various situations, such as while walking, riding a train, and lying down. It is known that the accuracy of pointing with the thumb is reduced [37] and the user's force resolution is reduced [47] while walking. In addition, the motion of the smartphone changes according to the user's body posture [13]. Therefore, body posture may impact performance in techniques that have a large smartphone movement. Moreover, because the situation of using a smartphone may also affect the performance of one-handed interaction techniques, we need to investigate it.

## 7 CONCLUSION

In this paper, to enable a user to perform all single-touch gestures to an unreachable area, we designed Force Cursor (FC) and Event Forward Cursor (EFC). FC is a technique to issue touch events (touch-down events when the force is increased and touch-up events when the force is decreased) at the cursor position using force. On the other hand, EFC is a cursor technique that consists of two steps of operation; the first is for determining the touch event position; the second is for performing single-touch gestures. Furthermore, we conducted the user study to investigate the single-touch gesture performance of three one-handed interaction techniques: contents shrinking technique (OM), FC, and EFC. From the results of the user study, although the time to perform a gesture and the stability of the smartphone did not vary greatly depending on the performing gesture, the success rate varied with the performing gesture: EFC had a high success rate regardless of the gesture, while OM and FC had a low success rate except for a tap. In addition, we found that both of the two techniques we designed to enable the user to interact with the smartphone with a stable grip.

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
