# OpenReview forum: "Design and Investigation of One-Handed Interaction Techniques for Single-Touch Gestures"
_graphicsinterface.org/Graphics_Interface/2021/Conference — Submitted to GI 2021_

### Official Review · AnonReviewer1 · 2021-01-13
**The paper is well written, except for a few shortcomings (listed in the review). I think with some minor revision, the paper is good to go.**

**Rating:** 7
**Confidence:** 5

**Review:**

# Summary of the paper
The paper introduces two one-handed interaction techniques for a large touchscreen mobile phone. The techniques are named FC and EFC, which utilize a remote cursor triggered by an edge gesture. Tap, double-tap, and drag could be performed by utilizing force (FC) and event forward (EFC), which should be the distinguishable contribution of this paper over existing works. However, the FC technique itself was already introduced in [14]. Therefore, the remaining contributions are the introduction of the EFC technique and their evaluations. I think the amount of new information on this paper is adequate to be published.

# Review
Overall, the paper is well written, except for a few shortcomings. I think with some minor revision, the paper is good to go.

## Validity of the idea
I empathize with the idea of allowing more gestures on the remote other than just a tap. Although the overall performance is less efficient than a direct touch and an existing solution (labeled OM in the paper), each new technique has its advantages: FC exhibited the most stable grip, and EFC exhibited the highest accuracy.

However, justifications of some design factors are missing: the choice of C-D gain (multiply by three), 1.0 sec dwelling time for "Drag Assistance" in FC technique. They seem to be set arbitrarily and would be beneficial to be justified with some references or research.

Also, nowadays, iOS and Android actively utilize edge gestures for other purposes, so a collision of gestures may happen. This should be discussed in the paper.

Performance-wise, I believe there will still be room for improvement. Especially regarding the last two paragraphs of section 5.4 -- compensation for subtle drift in force gesture and double-tapping is a common and widespread method in commercial products. I'd like to see the result again after patching this, where I expect a considerable improvement in time and accuracy of FC and EFC.

## The presentation of the statistical analysis results.
The way of presenting ANOVA and posthoc tests on sections 4.3.1, 4.3.2, and 4.3.3:
example) "Tukey's HSD test was also
significant (p < .05 between OM's DTap and OM's Swipe, OM's Drag and FC's DTap, FC's DTap and EFC's DTap, and EFC's Tap and EFC's Swipe, p < .01 between DT's Tap and DT's Swipe, DT's Tap and OM's Swipe, and DT's Swipe and OM's DTap, DT's Drag and FC's Swipe, DT's Drag and EFC's DTap, OM's Tap and OM's Swipe, and FC's DTap and FC's Swipe, and others p < 001."

The text is all monotonically listed just only with acronyms, "and", and commas, which makes it impossible to comprehend. I believe the information is there, but they are entirely unreadable.

Please consider a better presentation such as a table, bullet list, or annotation on the graph (e.g., https://www.researchgate.net/figure/Bar-graphs-of-post-hoc-tests-considering-the-area-A-mean-of-red-R-color-space-B_fig3_263430688). Maybe the interaction effects are hard to be presented in the table or graph. However, the current format is a disaster.

## Jerk and Angular Acceleration
The jerk and angular acceleration are time-series data. They cannot be calculated into a number in an obvious way, unlike the execution time and accuracy. The authors should clearly define how they drew out the numbers from the collected data.

## Discussion of the mix of DT and the technique
In reality, nobody will use any of the reachability compensation techniques (OM, FC, and EFC) on a target in the reachable area. So forced use of the technique on all the areas on the screen is unrealistic.

Therefore, a more realistic situation will be a mixed technique -- use DT for reachable targets and use one of the proposed techniques on distance targets. Allowing a user to choose between DT and a proposed technique will be a more externally valid experiment design. For example, [25] allowed the participants to choose one among direct touch and support techniques and presented a method preference on different screen areas. I think this should be discussed as an immediate follow-up study.

---

### Official Review · AnonReviewer2 · 2021-01-14
**Review of "Design and Investigation of One-Handed Interaction Techniques for Single-Touch Gestures"**

**Rating:** 5
**Confidence:** 5

**Review:**

Even though there is potential value in the investigation reported on here, the paper falls between two contributions (and falls short of both). One contribution would be in the new techniques themselves (Force Cursor and Event Forward Cursor), but for this to be a strong contribution, the paper would need to show that the techniques are widely applicable, and the evaluation would need to show that the new techniques are an improvement over the state of the art. However, the techniques seem limited in terms of how and when they can be used, and the evaluation does not show strong advantages for the new techniques (and in fact shows that they are significantly slower than what is already delivered on smartphones). Second, the contribution could be in the analysis of a new approach to one-handed use, namely the use of a cursor to avoid one of the limitations of direct touch; this is potentially interesting from a broader perspective (and for this contribution it would not be a problem to show that the cursor-based techniques are slower). However, the paper does not take on this question to any great degree, and so there is little that can be learned from the paper as a high-level comparison of general approaches. As a result, it is unclear whether the community will find much value in the paper as it stands now - the paper is currently written to introduce the new techniques (i.e., the first contribution type discussed above), but the evidence does not strongly argue for the techniques, and the second contribution (the high-level consideration of different input styles) is not developed enough in the current version.

Comments on the design of the techniques themselves. There appears to be some brittleness in the techniques' design, in terms of repeated actions with the same object. Both of the techniques turn off on liftoff (although EFC has uses hysteresis to allow double tap), and this means that for multiple actions with the same object, the user would need to start the technique multiple times. For example, consider an action series with a graphical object where the object is first selected, then dragged to a new location, and then double-clicked. This is easily done with direct touch, but with the proposed techniques, the user would need to invoke the cursor and move the cursor to the graphical object three times. I do understand that the authors were considering atomic gestures such as taps and swipes, but there are many situations where these atoms would be chained together in action sequences, and the design of the new techniques fails to take this into consideration.

The evaluation of the techniques shows that they are substantially slower than both the "ordinary" and the "shrink" conditions. This is a major problem for the techniques that essentially condemns them to being unusable from a design standpoint. The authors seem to have found a reasonable explanation for the slower operation - the use of a cursor instead of direct touch - but the authors do not dig deeper into this difference in a way that we learn much about the high-level differences between input approaches. Whatever the reason, being 1.5x to 2.0x slower than no treatment at all is a critical flaw - and even though there is a small improvement in accuracy and stability for the new techniques, the large time cost means that the techniques are unsuccessful. (I will note as well that the added time may well account for the improvements on other measures; in addition, the authors are overreaching when they state that DT's higher SUS score is due only to its familiarity - the high score could easily be because the DT technique was faster than the other techniques as well).

A couple of minor points:
- the swipe in from the bezel is already used in some smartphones (e.g., Pixel 4); the authors should consider interference between their gestures and existing gestures
- the dense presentation of follow-up results is difficult to parse and should be summarized

---

### Official Review · AnonReviewer3 · 2021-01-15
**Review of "Design and Investigation of One-Handed Interaction Techniques for Single-Touch Gestures"**

**Rating:** 5
**Confidence:** 5

**Review:**

The paper proposes two one-handed techniques for selecting unreachable targets: FC is a cursor technique based on the force, while EFC is another cursor technique that uses a two-step operation. The problem is well-motivated: the smartphones' screen size is increasing while the thumb reachability remains the same.

While I find the techniques potentially interesting, several drawbacks are not addressed by the authors that I think are critical:

Firstly, the description of techniques is unclear and hard to follow; however, watching the demo video several times helped to understand the general idea and the interaction design, which are simple. The images of Fig.1 are very helpful, yet the caption is distracting.

Secondly, several important decisions are not well-justified: e.g., the choice of the force threshold (1.8N) or the choice of the cursor-display ratio (being 3 times the finger movement distance), the choice of dwell time for the drag assistance. Conducting a pilot study or providing some references could help.

Finally, the authors highlighted some of the limitations; however, from the experimental results, it is not clear and hard to conclude that the proposed techniques can compete with existing methods. Due to some factors:

- Time and Accuracy tradeoffs:
EFC seems to be slightly more accurate (to 2~6%) using some gestures than DT, but still very slow compared to DT and OM ($1.5\times\sim2\times$ times). There is a chance that users could have reached better accuracy with DT when spending a little more time on precision/correction of the action and yet remain the fastest. This could be justified/rejected with the help of a longitudinal study and by recruiting more participants.

- Jerk and Angular Acceleration:
Since the main contribution of the paper is to show that the proposed techniques provide stable grasping with one hand, the discussion/elaboration on Jerk and Angular Acceleration data is required (e.g., how it was calculated, were there any observations, etc.). In 4.3.3, page 7, the authors assume that FC was the most "stable" due to continuous thumb touch on the screen, which does not necessarily mean stability (users can tap/double tap on the screen and result in shaking the phone and yet have stable grasp). It would be interesting to have a discussion section on how authors justify stability with the help of proposed metrics.

Overall, while I’m sympathetic towards the proposed techniques, the aforementioned issues prevent me from recommending the acceptance of this paper. As a side note, I would also be interested in seeing the comparison to iPhone's haptic touch/3D touch (where the onscreen keyboard turns into a virtual trackpad while holding the space bar). While it does not allow to perform “double-tap” or “drag”, it enables a continuous movement of the cursor through the text with high precision.



Minor things:

- Why did the authors decide to eliminate DT results from Tables 1 and 2?
- Page 2: Lochtefeld –> Löchtefeld et al.
- Page 4: use –> used
- Page 7: Therefore, We -> Therefore, we
- Subsection titles use modified font for 4.3.1, 4.3.2, 4.3.3, and 4.3.4. They should not: titles should use the same font face/size.
- The p values and other expressions involving mathematical symbols should be typesetted in math mode, e.g., using $. This will help with the readability of equations. (In sections 4.3.2, 4.3.3, and others)

---

### Meta-Review · Area_Chair1 · 2021-01-16

**Recommendation:** Reject
**Confidence:** 5

**Metareview:**

All reviewers appreciated the work but identified some major issues that must be addressed. We encourage the authors to consider re-submitting a revised version for the next review cycle.

---

### Decision · Program_Chairs · 2021-01-16

Reject